# Molecular diagnosis of intestinal protozoa in young adults and their pets in Colombia, South America

**Caterine Potes-Morales** [id][co], **Maria del Pilar Crespo-Ortiz** [co] *

Department of Microbiology, Section of Parasitology, Universidad del Valle, Cali, Colombia

[co] These authors contributed equally to this work.
* maria.crespo.ortiz@correounivalle.edu.co

**Data Availability Statement:** All relevant data are within the paper and its Supporting information files.

## Abstract

Intestinal parasitic infections have been considered a relevant public health problem due to the increased incidence worldwide. In developing countries, diarrhea and gastrointestinal symptoms cause impaired work capacity in adults and delayed rate growth in children. Enteric infections of unknown etiology can often lead to misdiagnosis, increased transmission, and morbidity. The aim of this study was to determine the prevalence of intestinal parasites in a young adult population and their pets. Stool samples from 139 university students and 44 companion animals were subjected to microscopy diagnosis using wet mounts, concentration by zinc sulphate flotation and staining techniques (Kinyoun and trichrome stain). Molecular diagnosis of protozoa was also performed by conventional PCR. The mean age was 24 years, 54% individuals were female, 46% were men, and 66% had at least one pet. The overall prevalence for at least one parasite was 74.8% and the rate of polyparasitism was 37.5%. Eighty-three patients (59.7%) were positive for *Blastocystis* spp., followed by *Cryptosporidium* spp. 24.5%, *Endolimax nana* 13.6%, *Entamoeba dispar/E. moshkovskii* 7.8% and *Giardia intestinalis* 1.4%. Molecular diagnosis substantially improved *Cryptosporidium* spp. and *Blastocystis* spp. detection and allowed to distinguish *E. histolytica* from commensals in the *Entamoeba* complex. Student's pets were also examined for parasitism. Samples from 27 dogs, 15 cats, one rabbit and one hen were analyzed, and parasites were detected in 30 (68.2%) as follows: *Cryptosporidium* spp. (24) *Giardia* spp. (4), hookworm (3), *Endolimax nana* (2) and *Toxoplasma gondii* (1). Overall, university students showed high prevalence of parasitism and polyparasitism suggesting exposure to parasite infected animals and contaminated environments. *Cryptosporidium* spp. was the predominant pathogen in human and domestic animals, and it was only detected by PCR, pointing out the need for sensitive tests in diagnosis and surveillance. Control strategies to prevent the effects of parasitic infections in young population should consider pets as reservoirs and transmission source.

**Funding:** This research was financially supported by the Internal Grant Scheme 2020 (CI1920) awarded to CPM and MPC by the Universidad del Valle, Cali, Colombia. The funder had no role in study design, data collection and analysis, decision to publish, or preparation of the manuscript.

**Competing interests:** The authors have declared that no competing interests exist.

## Introduction

In the last decade, intestinal parasitic infections have been considered a relevant public health problem due to the increased incidence worldwide. Low and middle-income countries are mainly affected as parasite infections are associated with poor sanitation, lack or poor quality of health care access and limited diagnosis [1, 2]. In high income countries, several associated factors such as human migration, travel, animal exposure and immunosuppressed populations are responsible for the increase in infections [3–5]. Globally, 4 billion people could be affected by intestinal helminths and protozoa [6]. More than 1.45 billion people (24% of the world's population) are infected with soil-transmitted helminths (STH) mainly by *Ascaris lumbricoides*, *Trichuris trichiura* and hookworms. STH prevail in Africa, the Americas, China and East Asia, with a socioeconomical impact of 5.2 million Disability Adjusted Life Years (DALYs) [7] and more than 3 million DALYs globally [8]. The metric DALY represents the sum of years lost due to premature mortality and the years lived with disability due to a health condition [9]. This indicator has been used to measure the impact of intestinal parasite infections to address primary prevention programs [10].

Although less frequent than STH, intestinal protozoa are relevant contributors to the diarrheal disease worldwide causing 357 million cases, 33.900 deaths and 2.94 million DALYs. At least 67.2 million cases have been associated with foodborne transmission due to high environmental contamination and poor sanitation [1]. Foodborne infections associated to pathogen protozoa are up to 28 million, mainly caused by *Entamoeba histolytica* and *Giardia intestinalis*, whereas up to 8.5 million are caused by *Cryptosporidium* spp. Associated mortality has been reported for *Cryptosporidium* spp. and *E. histolytica* infections [1] with higher impact in immunosuppressed population and low income countries [11]. Other protozoa frequently found but with controversial pathogenicity are *Dientamoeba fragilis* and *Blastocystis* spp. *Dientamoeba fragilis* has been selectively reported in developed countries ranging from to 0.2 to 82%, this variability depends on the geographical area, population group, study design and diagnostic procedures [12–14]. *Blastocystis* spp. has been found colonizing over 1 billion people around the world and its role in host microbiota interactions and other gut health conditions remains to be elucidated [15, 16].

Parasite prevalence may vary by geographical area, contamination levels, environmental conditions, and detection systems. According to some estimates, 45% of the population in developing countries from the Americas is infected, however parasite surveys have shown high variability across several countries, as in some cases, data are limited to small geographical areas or populations, and deworming policies and sanitation conditions may be different [17]. A national survey in school children in Colombia revealed a prevalence of 17% for *E.histolytica/E. dispar/ E. moshkovskii*, 15% for *G. intestinalis* and 0.5% for *Cryptosporidium* spp. with more than half the population (58%) infected with the commensal genus *Blastocystis* [18]. Most studies have been focused on children and using microscopy detection but information from other potentially vulnerable populations is still scarce.

Parasite infections commonly cause gastrointestinal symptoms and acute or chronic diarrhea leading to impaired work capacity in adults and delayed rate growth in children [2]. The vast majority of enteric infections remains with unknown etiology leading to misdiagnosis, and increased transmission and morbidity, therefore surveillance using improved diagnostic tools is pivotal to address control strategies. In most low-middle income countries, parasite detection is mainly performed by microscopy-based methods which are simple and low cost but fully rely on morphology. Microscopy has intrinsic limitations such as low or variable sensitivity and lack of species differentiation. Alternatively, molecular diagnosis using DNA

amplification methods has shown increased sensitivity and specificity which also offer multiplex formats and make feasible further genotype analysis [19, 20].

In developing countries, better surveillance and improved routine diagnosis tests are required to understand the clinical effects of parasite infections in all affected populations and to reduce transmission. The aim of this study was to determine the prevalence of intestinal parasites in a student young adult population using conventional microscopy and molecular diagnosis of relevant protozoa. We also aimed to explore the factors that may favor the parasite spread and particularly, the potential role of animal exposure.

## Materials and methods

### Ethics statement

The study protocol was approved by the Universidad del Valle Ethics Committee (Approval No.: 031/CNES/2010). Prior to the study, written informed consent from all participants was obtained.

### Population study

A cross sectional study was conducted in a higher education institution, a public and research university in Cali, Colombia. Student population is predominantly from low and low-middle income backgrounds and comes from Cali and the nearest cities from the Departments of Valle del Cauca, Cauca, and Nariño. Cali is the third most populous city in southwest Colombia (3˚27′00′′N, 76˚32′00′′W) with dry-summer tropical climate and average annual precipitation between 900 to 1,800 mm and 25˚C temperature. A total of 139 students were enrolled in the study from August 2021 to January 2022. Students were recruited according with the following criteria: current enrollment at the university, age 18–40 and no parasite treatment for at least 6 months prior to the study. After an introductory talk to the research, the participants signed the consent form and then filled a self-administered questionnaire with socio-demographic, exposure, and clinical variables such as: gastrointestinal symptoms at the time of sampling, past or current health conditions, medications and current or previous parasite infections. All the data were included for analysis.

Fecal samples were obtained and processed using conventional microscopy-based techniques and then preserved in Shaudinn´s fixative for trichrome stain. The remainder of each sample was kept at -80˚C for further molecular analysis of enteric protozoa.

### Microscopy

After macroscopic inspection (color and consistency), stools were examined under microscope using saline and lugol´s solution to identify trophozoites, cysts, helminth ova and larvae. Simultaneously, samples were subjected to concentration by the zinc sulphate flotation method. Briefly, one gram of feces was thoroughly mixed and washed twice, 7 mL of zinc sulphate (specific gravity 1.18) was added and centrifuged for 1000g x 2 minutes. The upper biofilm was examined under microscope using saline and lugol´s solution [21]. Parasite load was estimated in positive samples (wet mounts) and defined by the number of parasites per 100 low power fields (lpfs) as follows: scarce (1–10), low (11–25) moderate (26–50) or heavy (> 50) parasite load.

Additionally, fecal smears were prepared and subjected to modified acid -fast stain (Kinyoun) for detection of coccidian parasites [21]. For a better differentiation of parasite structures and quality control of microscopy, trichrome stains were also performed. Briefly, fecal smears were prepared and transferred for three minutes into D'Antoni iodine mixed with

70% ethanol. Then, the slides were placed in 70% ethanol for three minutes before being stained in the trichrome working solution (26 g/L) for 12 minutes and rinsed in 90% and 95% ethanol. Finally, the smears were placed in two changes of carbol-xylol solution and mounted using Permount Montage Medium (Fisher Chemical, TM) [21]. The smears were examined under a research microscope, Zeiss Axio imager A2 with an image analysis software (Zen Lite version 3.1).

## Molecular assays

**DNA extraction.** Stool samples were pretreated to improve DNA extraction. Briefly, 1 g of fecal sample was washed in sterile water and suspended in 250 μL of lysis buffer (0.15 M NaCl, 0.1 M EDTA, 0.5% sodium dodecyl sulphate, SDS) and vortexed for 20 minutes before being frozen at -80˚C overnight. Then, the samples were thawed and heated at 95˚C x 10 minutes before adding 3 μL of proteinase K (22 mg/mL) and then incubated for 10 minutes at 56˚C [22]. The homogenized samples were subjected to total DNA extraction using a fecal DNA extraction kit (IBI Scientific) according to the manufacturer´s recommendations.

**PCR amplification.** A conventional monoplex PCR was performed using specific primers to target genes from the pathogen protozoa *E. histolytica*, *G. intestinalis*, *Cryptosporidium* spp. [23] and *D. fragilis* [24] and for the commensal *Blastocystis* spp. [25]. PCR for *Entamoeba dispar* [26] (Table 1) was performed to identify *Entamoeba* species from the *Entamoeba* complex (*E. histolytica/E. dispar/E. moshkovskii*). PCR reaction mixtures were prepared to a final volume of 25 μL by adding 5 μL of DNA template, 0.2 μM of each primer, 0.2 mM dNTP mix and 1.25 U Taq polymerase. Primers, thermocycling and experimental conditions are shown in Table 1. Negative controls (no template) and positive DNA controls were included in each run. Parasitic DNA controls were: *E. histolytica* DNA from a clinical sample (kindly donated by Dr. Christen Rune Stensvold, Statens Serum Institut, Copenhagen, Denmark), *G. intestinalis* and *Blastocystis* spp. DNA from the laboratory collection, *D. fragilis* synthetic DNA (Microbiologics, Helix Elite) and *Cryptosporidium* spp. (BD MAX™ Enteric Parasite Control Panel, Microbiologics). Potential inhibition of PCR reactions was tested by amplification of known DNA in negative samples. PCR products were visualized in agarose gels using a Gel Doc system (LED FastGene FAS-DIGI PRO, NIPPON Genetics Europe).

Selected PCR amplified products were subjected to Sanger sequencing using an ABI 3730XL sequencer (Macrogen® Corp., Seoul, South Korea). The sequences were processed using Bioedit v 7.2.5 before alignment in CLUSTALW (MEGA Xv10.2.6). The BLAST tool (http://blast.ncbi.nlm.nih.gov/Blast.cgi) was used to determine the homology among sequences deposited in the National Center for Biotechnology Information (NCBI).

## Statistical analysis

The data were collected in an Excel database (Microsoft office Excel 365, version 2204) and exported to the Statistical Package for the Social Sciences (SPSS, version 27) for analysis. Data were also computed using Epi info v 7.2.5. Univariate analysis was performed by frequency tables and means. Associations among demographic or behavioral variables and parasite infection were assessed using contingence tables and odds ratios were calculated with a 95% confidence interval. Chi square or Fisher´s exact tests were used for significance level, p values <0.05 were considered significant. Diagnostic techniques were compared using the Cohen´s kappa coefficient (κ) taking as reference the combined results of all techniques for each parasite. Interpretation of kappa values was as follows: < 0.20 none to slight, 0.21–0.40 fair, 0.41–0.60 moderate, 0.61–0.80 substantial, and 0.81–1.00 almost total agreement [27]. Multivariate logistic regression modeling was also used to identify the socio-demographic and behavioral

**Table 1. Primer and DNA amplification conditions.**

| Parasite | Primers | Sequence | Target gene | Size bp | Thermocycling conditions | |
|---|---|---|---|---|---|---|
| | | | | | #Cycles | Thermal settings |
| *E. histolytica* [23] | EHCP8-S1 | ATTTGTTAAGTATTGTAAATGGG | CP8[a] | 605 | 1 | 94°Cx 5 min |
| | EHCP8-As1 | ATTGTAACCTTTCATTGTAACAT | | | 35 | 94°Cx 1 min |
| | | | | | | 55°Cx 1 min |
| | | | | | | 72°Cx 30s |
| | | | | | 1 | 72°C x 7 min |
| *G. intestinalis* [23] | GLCP6-S1 | AATCTGTTGACTTAAGGGAGTA | CP6[b] | 463 | 1 | 94°Cx5min |
| | GLCP6-As1 | ATTGAGTCATTATAGGGATTGT | | | 35 | 94°Cx 1 min |
| | | | | | | 55°Cx 1 min |
| | | | | | | 72°Cx 30s |
| | | | | | 1 | 72°C x 7 min |
| *Cryptosporidium* spp. [23] | CRY18s-S1 | TAAACGGTAGGGTATTGGCCT | SSU rRNA | 240 | 1 | 94°Cx5min |
| | CRY18s-As1 | CAGACTTGCCCTCCAATTGATA | | | 35 | 94°Cx 1 min |
| | | | | | | 60°Cx 1 min |
| | | | | | | 72°Cx 30s |
| | | | | | 1 | 72°C x 7 min |
| *D. fragilis* [24] | DF400 | TATCGGAGGTGGTAATGACC | 18S rRNA | 850 | 1 | 94°Cx3min |
| | DF1250 | CATCTTCCTCCTGCTTAGACG | | | 30 | 94°Cx 1 min |
| | | | | | | 60°Cx 1.5min |
| | | | | | | 72°Cx 2 min |
| | | | | | 1 | 72°C x 5 min |
| *Blastocystis* spp. [25] | bl1400ForC | GGAATCCTCTTAGAGGGACACTATACAT | SSU rRNA | 310 | 1 | 94°Cx7min |
| | bl1710RevC | TTACTAAAATCCAAAGTGTTCATCGGAC | | | 35 | 94°Cx 1 min |
| | | | | | | 60°Cx 1 min |
| | | | | | | 72°Cx 1 min |
| | | | | | 1 | 72°C x 7 min |
| *E. dispar* [26] | ED-1 | TCTAATTTCGATTAGAACTCT | 18S rRNA | 174 | 1 | 96°Cx 2min |
| | ED-2 | TCCCTACCTATTAGACATAGC | | | 30 | 92°Cx 1 min |
| | | | | | | 51°Cx 1 min |
| | | | | | | 72°Cx 1.5 min |
| | | | | | 1 | 72°C x 7 min |

[a] CP8 Cysteine protease 8,

[b] CP6 Cysteine protease 6

variables associated with parasite infection. Over 30 variables in the univariate analysis, including those recognized as risk factors for parasite infection, were entered into a backward stepwise (Likelihood ratio) logistic regression model in SPSS v 27.

# Results

## Socio-demographic characteristics of the study population

A total of 139 university students were included, 75 (54%) were female and 64 (46%) were male (male/female ratio 0.85) the mean age was 24 years old. Most of the study population (89, 64%) was among 18 and 24 years and coming from families considered in the low or middle low-income range. Most participants reported habits such as tap water drinking (115, 82.7%), fruit and vegetables consumption (122, 87.8% and 109, 78.4% respectively) and relevant animal

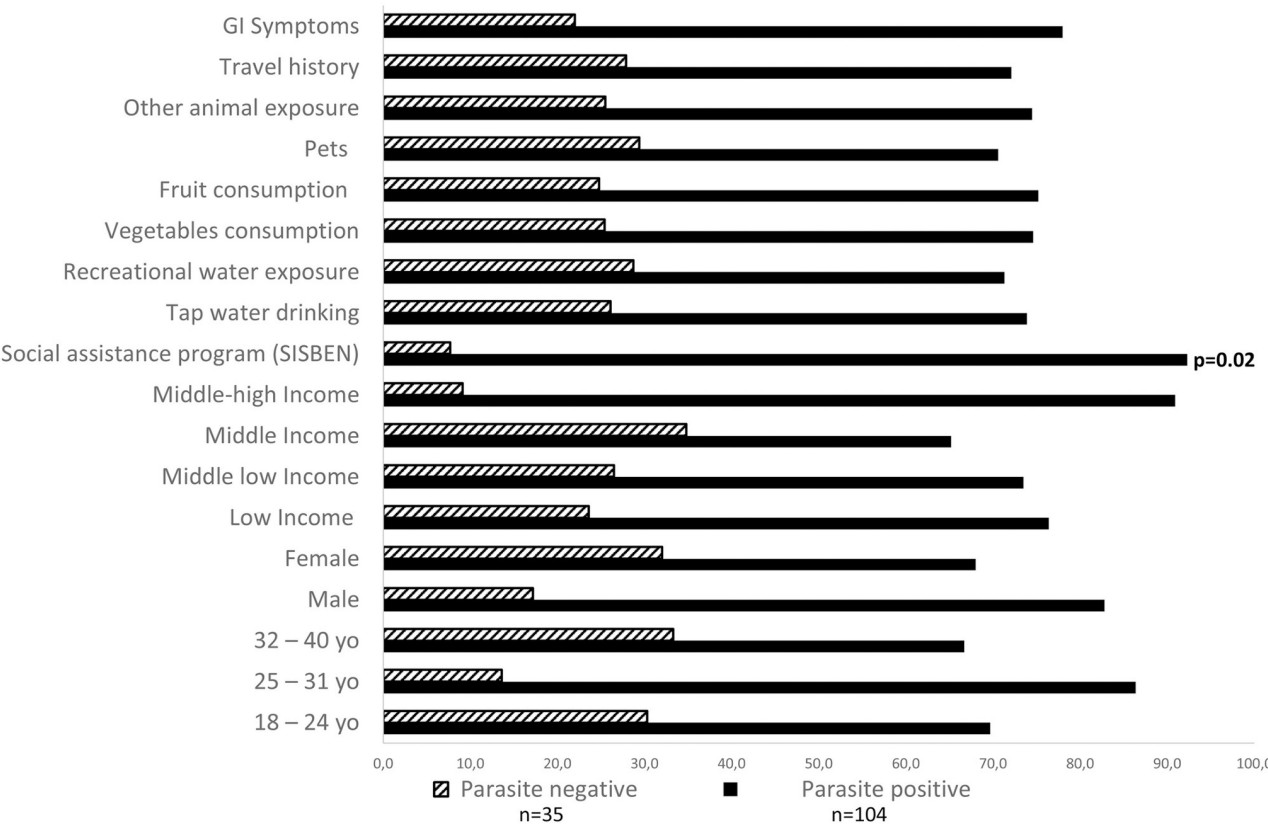

**Fig 1. Population of study by parasite infection.** Baseline characteristics of young adults by parasite infection status are shown (n = 139). GI: gastrointestinal.

exposure including pets (92, 66.2%) or other animals (55, 39.6%). Fifty (36%) students reported gastrointestinal symptoms at the time of sampling. A description of the study population by parasite infection status is shown in Fig 1. Parasite infection was associated with participants under the health care assistance program (SISBEN), this program is a vulnerability assessment and identification system of beneficiaries for social assistance in Colombia (Fig 1 and Table 2). Although no significant associations were found among overall parasitism and any other sociodemographic, behavioral, academic and clinical factors, those aged 25–31 years showed to be more likely infected (p = 0.054 95% CI 1.05–7.30, Table 2). Likewise, according to the multivariate logistic regression, the only association with parasite infections in this population was being enrolled under the health care assistance program (SISBEN) (p = 0.014, 95% CI: 1.51–42.90), supporting the results obtained in the bivariate analysis, other factors included in the regression model did not show any relevant interactions or significant association.

## Parasite detection by microscopy

Seventy-one (51.1%) samples were formed stools, 62/139 (44.6%) were semiformed and 2/139 (1.4%) were watery, no bloody fecal samples were observed. Mucus was observed in two samples and leucocytes were not seen.

Intestinal parasites were found in 63/139 (45.3%) students using microscopy techniques: 41/139 (29.5%) were positive for direct wet mount whereas 54/128 (42.2%) were seen by

**Table 2. Sociodemographic characteristics of young adults and parasitism associated factors.**

| Variable | Total | Parasite positive | Parasite negative | P value | 95%CI |
|---|---|---|---|---|---|
| | n = 139 | n = 104 | n = 35 | | |
| | No. (%) | No (%) | No (%) | | |
| Age (years) | | | | | |
| 18–24 | 89 (64.0) | 62 (69.7) | 27 (30.3) | 0.09 | 0.18–1.06 |
| 25–31 | 44 (31.7) | 38 (86.4) | 6 (13.6) | 0.054 | 1.05–7.30 |
| 32–40 | 6 (4.3) | 4 (66.7) | 2 (33.3) | 0.64 | 0.09–7.63 |
| Gender | | | | | |
| Male | 64 (46.0) | 53 (82.8) | 11 (17.2) | | |
| Female | 75 (54.0) | 51 (68) | 24 (32) | 0.07 | 1.00–5.10 |
| Income [a] | | | | | |
| Low | 55 (39.9) | 42 (76.4) | 13 (23.6) | 0.85 | 0.52–2.56 |
| Middle- Low | 49 (35.5) | 36 (73.5) | 13 (26.5) | 0.97 | 0.41–2.01 |
| Middle | 23 (16.7) | 15 (65.2) | 8 (34.8) | 0.38 | 0.22–1.50 |
| Middle-high | 11 (8) | 10 (90.9) | 1 (9.1) | 0.28 | 0.48–168 |
| Health care provider | | | | | |
| Social assistance program (SISBEN) | 26(18.7) | 24 (92.3) | 2 (7.7) | 0.02[b] | 1.11–45.2 |
| Source of drinking water | | | | | |
| Tap | 115(82.7) | 85 (73.9) | 30 (26.1) | 0.77 | 0.25–2.17 |
| Recreational water exposure | 94(67.6) | 67 (71.3) | 27 (28.7) | 0.23 | 0.22–1.30 |
| Food habits (consumption) | | | | | |
| Vegetables | 122 (87.8) | 91 (74.6) | 31 (25.4) | 1.00 | 0.19–3.21 |
| Fruits | 109 (78.4) | 82 (75.2) | 27 (24.8) | 1.00 | 0.14–2.76 |
| Animal exposure | | | | | |
| Pets | 92 (66.2) | 65 (70.7) | 27 (29.4) | 0.16 | 0.20–1.19 |
| Other animals | 55(39.6) | 41 (74.5) | 14 (25.5) | 1.00 | 0.44–2.15 |
| Travel history | 104(74.8) | 75 (72.1) | 29 (27.9) | 0.29 | 0.20–1.42 |
| GI Symptoms | 50 (36.0) | 39 (78) | 11 (22) | 0.65 | 0.57–2.96 |

GI: Gastrointestinal.

[a]Economic stratification is based on living standards and housing conditions.

[b]P Fisher test: statistically significant.

trichrome stain (Table 3). From the negative samples for wet mount, which is the routine diagnosis test, one was detected by flotation concentration and 21 in trichrome stain. Results from the flotation technique were like those from direct wet mount except for the commensals *Blastocystis* spp. and *Chilomastix mesnili* which were not recovered using this method.

Combining all microscopy methods, *Blastocystis* spp. was the parasite most frequently found 47/139 (33.8%) followed by *Endolimax. nana* 19/139 (13.6%), *E. histolytica/E. dispar/E. moshkovskii* 10/139 (7.2%), *Entamoeba coli* 7/139 (5%), *G. intestinalis* 2/139 (1.4%), *Entamoeba hartmanni* 2/139 (1.4%), *Iodamoeba bütschlii* 1/139 (0.7%), *Chilomastix mesnili* 1/139 (0.7%) and *D. fragilis* 1/139 (0.7%) (Fig 2). Helminths eggs and *Cryptosporidium* spp. oocysts were no detected by microscopy.

Overall, the presumptive frequency of pathogenic protozoa by microscopy was 8.6% (12/139). Protozoan loads were mostly low (54.2%), commensal parasites such as *E. nana* and *E. coli* showed higher cysts loads (33% and 50% respectively) whereas *Blastocystis* fecal loads were low (61.5%).

**Table 3. Prevalence of intestinal parasites by microscopy and molecular methods.**

| | Diagnosis test | | | | Overall prevalence |
|---|---|---|---|---|---|
| | Wet mount | Zinc flotation | Trichrome stain | PCR | |
| | n = 139 | n = 138 | n = 128 | n = 139 | |
| | N (%) | N (%) | N (%) | N (%) | |
| **Protozoa** | 41 (29.5) | 23 (18.1) | 54 (42.2) | 91 (65.5) | 104 (74.8) |
| *Blastocystis* spp | 26 (18.7) | 0 (0) | 47 (36.7) | 69 (49.6) | 83 (59.7) |
| *Endolimax nana* | 15 (10.8) | 16 (11.6) | 17 (13.3) | N/A | 19 (13.6) |
| *Entamoeba* Complex | 6 (4.3) | 7 (5.1) | 10 (7.8) | 0 (0) | 10 (7.8) [a] |
| *Entamoeba coli* | 5 (3.6) | 6 (4.3) | 3 (2.3) | N/A | 7 (5.0) |
| *Entamoeba hartmanni* | 1 (0.7) | 1 (0.7) | 2 (1.6) | N/A | 2 (1.4) |
| *Iodamoeba bütschlii* | 1 (0.7) | 0 (0.0) | 1 (0.8) | N/A | 1 (0,7) |
| *Chilomastix mesnili* | 1 (0.7) | 0 (0.0) | 0 (0.0) | N/A | 1 (0.7) |
| *Giardia intestinalis* | 1(0.7) | 2 (1.4) | 2 (1.6) | 0 (0.0) | 2 (1.4) |
| *Dientamoeba fragilis* | N/A | N/A | 1 (0.8) | 0 (0.0) | 1 (0.7) |
| *Cryptosporidium* spp. | N/A | N/A | Kinyoun[b] | 34 (24.5) | 34 (24.5) |

N/A: Not applicable. Parasite identification by microscopy was based on typical morphology features [21].

[a]*Entamoeba* complex presumptively indicates *E. histolytica*/*E. dispar*/*E. moshkovskii* using microscopy. In this work *E. histolytica* was not detected by PCR. *E. dispar* was detected in 4 out of 10 positive samples for *Entamoeba* complex.

[b]All samples were Kinyoun negative.

## Molecular detection of parasites

Ninety-one (65.5%) participants were positive for parasites by molecular diagnosis. For the five parasites assessed, PCR only identified two types: *Blastocystis* spp. 69/139, 49.6% and *Cryptosporidium* spp. 34/139, 24.5%, eighteen (18/91, 19.8%) were positive for both. No DNA from *E. histolytica*, *G. intestinalis* or *D. fragilis* was amplified. Ten positive samples for *E. histolytica/dispar/moshkovskii* by microscopy were negative for *E. histolytica* by PCR. These samples were subjected to PCR for *E. dispar* and only four of them were positive. Results from microscopy and molecular diagnosis for each parasite are shown in Table 3.

Performance of each technique by parasite indicates that PCR was more effective for *Blastocystis* spp. and *Cryptosporidium* spp. than microscopy, with an overall agreement of 61.2% (Cohen´s k = 0.24, 95% CI: 0.097–0.369, fair agreement). In total, 83 stool samples were positive for *Blastocystis* spp. PCR positive samples were 49.6% (69) whereas 18.7% (26) were positive by wet mount and 33.8% (47) by any microscopy method. As shown in Fig 3, thirty-eight samples were positive by both PCR and microscopy, 31 were only positive by PCR and 14 were detected by microscopy using trichrome stain (69% agreement, Cohen´s k = 0.38, 95% CI: 0.232–0.528, fair agreement). For *Cryptosporidium* spp. all 34 cases were detected only by PCR (24.5% vs 0% for microscopy) with 0% positive agreement. However, the only two positive samples of *G. intestinalis* and one presumptive *D. fragilis* could not be confirmed by PCR. When comparing both techniques to identify the presence of any parasite in the samples, the prevalence estimated by conventional PCR was 65.5% (91) in contrast to 45.3% (63) with any microscopy technique.

Thirteen out of 34 positive samples, including the *Cryptosporidium* spp. and *E. dispar* DNA amplicons were subjected to Sanger sequencing. BLAST analyses showed >96% homology of *Cryptosporidium* amplicons under accession numbers ON668107 to ON668114 to the available GenBank sequences. One was identified as *C. parvum* and one as *C. felis*. The *E. dispar*

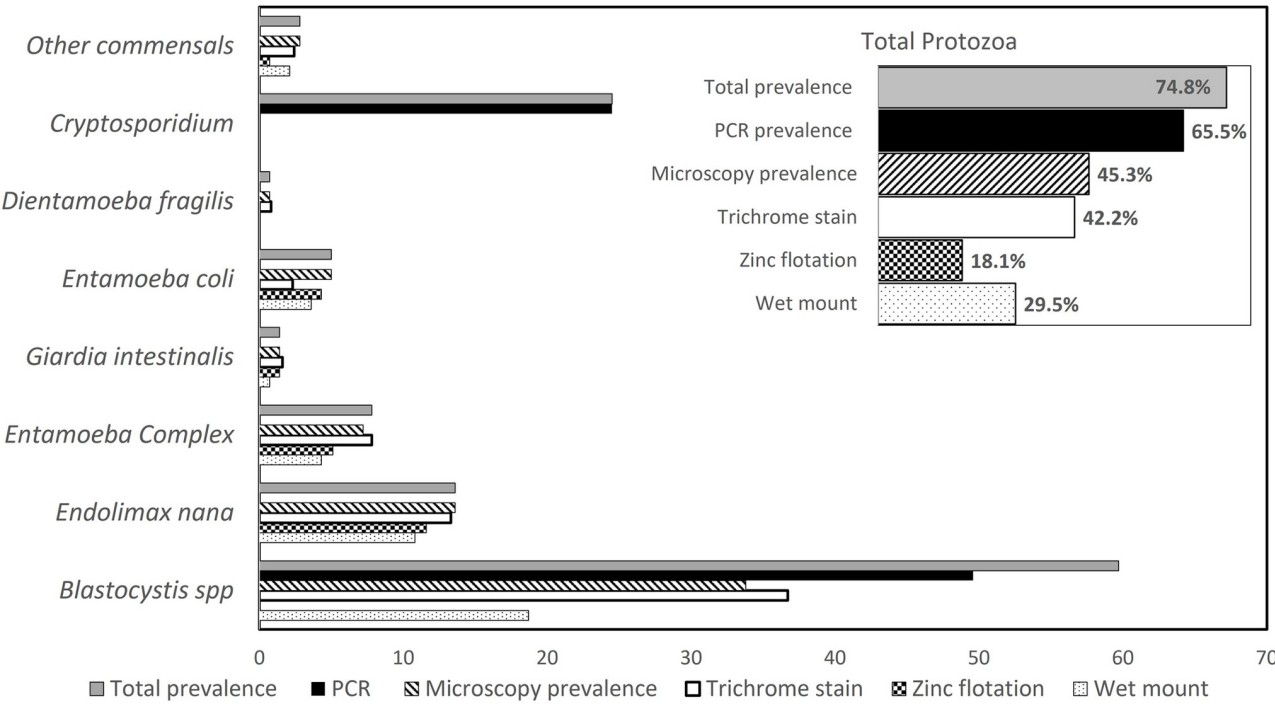

**Fig 2. Parasite detection by microscopy methods and PCR.** Prevalence of intestinal parasites according to each microscopy method and PCR. None of the cases considered as *Entamoeba* complex was positive for *E. histolytica* by PCR. Other commensals included: *E. hartmanni*, *I. bütschlii* and *C. mesnili*. Overall prevalence by method is shown in the inset.

sequence with accession number ON668115 was 100% identical to GenBank sequences of *E. dispar* under accession number MT250839.1.

## Single and multiple parasite infections

The overall parasite prevalence, which was determined by combining results from both microscopy and PCR techniques, was 74.8% (104/139). *Blastocystis* spp. 59.7% (83/139),

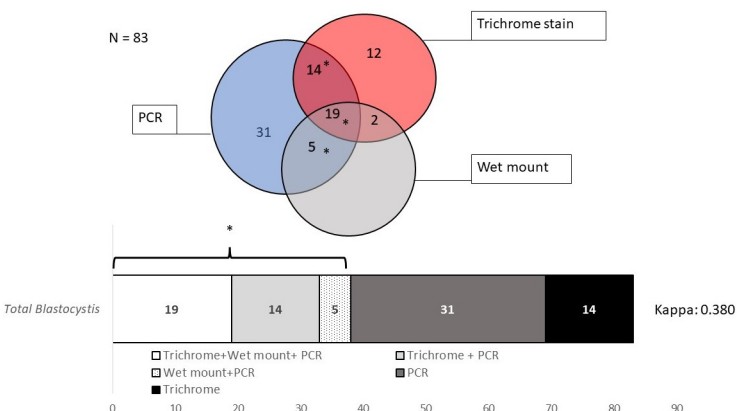

**Fig 3. Microscopy and PCR concordance for *Blastocystis* spp.** Agreement analysis is shown in overlapping circles. Positive agreement between any microscopy test and PCR is indicated by the asterisk.

*Cryptosporidium* spp. 24.5% (34/139), *E. nana* 13.6% (19/139), *E. dispar/E. moshkowskii* 7.8% (10/139), *E. coli* 5% (7/139) and other protozoa 4.2% (Table 3 and Fig 2). For specific parasite associations, it was seen that infection by *Blastocystis* spp. was frequently found in participants reporting fruit consumption (p = 0.0018, 95% CI 1.72–9.56) in contrast, taking home-prepared meals showed a protective effect (Fisher´s p = 0.038, 95% CI 0.07–0.93). No demographic or clinical factors were associated with *Cryptosporidium* spp. infections.

Monoparasitism was more prevalent in the parasite infected individuals (62.5%, 65/104) whereas polyparasitism was observed in 37.5% (39/104) and distributed as follows: 25/39 (64.1%) double, 11/39 (28.2%) triple, 1/39 (2.6%) quadruple and 2/39 (5.2%) quintuple infections, respectively. Polyparasitism was observed among commensals and pathogens but infections with more than one pathogen were not found. The most frequent coinfections were: *Blastocystis* spp. in combination with *Cryptosporidium* spp. 30.8% (12/39) or *E. nana* 20.5% (8/ 39) which were commonly identified in double infections. The most frequent triple infections observed included *Blastocystis* spp., *E. nana* and *Cryptosporidium* spp. (Fig 4). It was found that pet owners were less prone to have multiple parasite infections (p = 0.033 OR: 0.19–0.87). No further associations with polyparasitism were seen.

## Animal exposure

Participants with pets were asked to bring samples from their companion animals; a total of 31 out of 92 (33.7%) owners submitted fecal pet samples. Forty-four samples were subjected to microscopy detection of parasites using the same combined methodology as for the owners. PCR for *Cryptosporidium* spp. was also performed. Stools from 27 dogs, 15 cats, one rabbit and one hen were examined. Thirty (68.2%) pets were positive for intestinal parasites, the

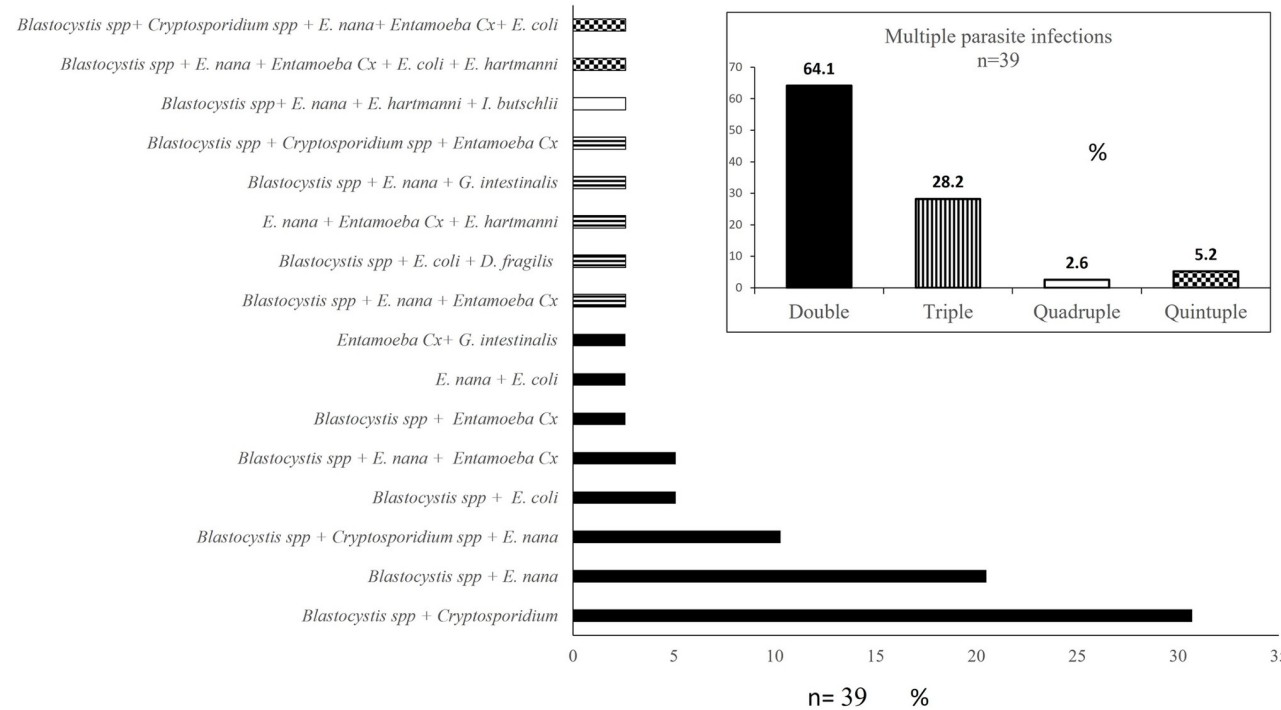

**Fig 4. Multiple parasite infections.** Parasite combinations distributed as: double infections (solid bars), triple infections (striped bars), quadruple infections (unfilled bar) and quintuple infections (squared bars) are shown. *Entamoeba* Cx: *Entamoeba* complex (i.e *E. dispar/E. moshkovskii* as *E. histolytica* was not detected in this work). Frequency of coinfections is shown in the inset.

**Table 4. Intestinal parasites in domestic dogs and cats.**

|  | N (%) |
|---|---|
| **Dogs n = 27** |  |
| **Total positive dogs** | 15 (55.6) |
| *Cryptosporidium* spp. | 14 (51.9) |
| Hookworm | 3 (11.1) |
| *Giardia* spp. | 1 (3.7) |
| **Multiple infections** |  |
| *Giardia* spp. + Hookworm | 1 (3.7) |
| Hookworm + *Cryptosporidium* spp. | 1 (3.7) |
| **Cats n = 15** |  |
| **Total positive cats** | 10 (66.7) |
| *Cryptosporidium* spp. | 8 (53.3) |
| *Giardia* spp. | 3 (20.0) |
| *E. nana* | 2 (13.3) |
| *Toxoplasma gondii* | 1 (6.6) |
| **Multiple infections** |  |
| *Giardia* spp. + *E. nana* | 2 (13.3) |

remaining pets from 11 owners were negative. Cats showed more parasite infections than dogs, *Cryptosporidium* spp. was the most prevalent parasite in both species (8/15, 53.3% in cats, 14/27, 51.9% in dogs) followed by hookworms in dogs and *Giardia* spp. in cats (Table 4).

Human pathogens were also found in pets including one cat with *Toxoplasma gondii* infection. The pets with parasitic infections (30) belonged to 20 owners, the parasites found in pets and the parasitic status of owners are shown in Table 5. Seven out of 20 (35%) owners shared the same parasite type with their pets and 8 out of 20 (40%) owned more than one pet. Remarkably, it was seen that several pets in a household usually had the same parasite type (Table 5).

## Discussion

Despite all the control measurements, intestinal parasites infections remain high worldwide. Considerable amount of research has focused on the most vulnerable populations but only few studies have explored the magnitude of parasite infections and associated factors in young adult population. The present work is the first study using combined microscopy and molecular diagnosis and exploring the animal exposure in this population. This survey revealed that 74.8% of the student young adults were infected with at least one parasite. Parasite infections were associated with those beneficiaries of the social assistance program (SISBEN), a Colombian system to identify vulnerable population in need. Here, the prevalence of parasite infection was higher when compared to previous studies in student young adults from several low-middle income countries ranging from 9% to 45.6% (Ethiopia 45.6% [28], Nigeria 9.3% [29], Bangladesh (23.1%) [30] and Iran 11.9% [31]) Table 6. Furthermore, two studies in Colombia have also shown high parasite prevalence in a similar population (81–83.4%) [32, 33]. No associations were found in gender, age or participant´s procedence, this is in contrast with studies in Ethiopia where parasites were more frequently found in males [28], rural residents, married students and those enrolled longer than one year [34]. The majority of studies conducted in young adults has been performed using only conventional microscopy except those in Mexican students focused on molecular diagnosis and typing of *Blastocystis* spp. [15, 35].

**Table 5. Positive pets and parasitic status of their owners.**

| | Parasite positive pets | Parasitic status of owner |
|---|---|---|
| | n = 30 | n = 20 |
| Type | Parasite | Parasite |
| Dog | *Cryptosporidium* spp. | *Cryptosporidium* spp. |
| Dog | *Cryptosporidium* spp. | *Cryptosporidium* spp., *Blastocystis* spp. |
| Dog #1 | *Cryptosporidium* spp., Hookworm | *Blastocystis* spp. |
| Dog #2 | *Cryptosporidium* spp. | |
| Dog | *Cryptosporidium* spp. | *Blastocystis* spp. |
| Cat | *Giardia* spp. | |
| Cat | *Toxoplasma gondii* | *Blastocystis* spp., *E. coli* |
| Dog | *Cryptosporidium* spp. | *Cryptosporidium* spp., *Blastocystis* spp., *E. nana* |
| Dog | *Cryptosporidium* spp. | *Cryptosporidium* spp., *Blastocystis* spp., *E. nana* |
| Dog | *Giardia* spp., Hookworm | *Blastocystis* spp., *E. nana* |
| Dog | *Cryptosporidium* spp. | *Cryptosporidium* spp. |
| Cat #1 | *E. nana*, *Giardia* spp. | *Blastocystis* spp. |
| Cat #2 | *E. nana*, *Giardia* spp. | |
| Dog | *Cryptosporidium* spp. | *Cryptosporidium felis*[a] *Blastocystis* spp., *E. nana.* |
| Rabbit | *Cryptosporidium* spp. | |
| Hen | *Cryptosporidium* spp. | |
| Dog | Hookworm | *Blastocystis* spp. |
| Dog | *Cryptosporidium* spp. | Negative for parasites |
| Cat #1 | *Cryptosporidium* spp. | *Cryptosporidium* spp. |
| Cat #2 | *Cryptosporidium* spp. | |
| Dog #1 | *Cryptosporidium* spp. | Negative for parasites |
| Dog #2 | *Cryptosporidium* spp. | |
| Cat | *Cryptosporidium* spp. | |
| Cat #1 | *Cryptosporidium* spp. | *E. dispar* |
| Cat #2 | *Cryptosporidium* spp. | |
| Cat | *Cryptosporidium* spp. | Negative for parasites |
| Dog | *Cryptosporidium* spp. | *Blastocystis* spp. |
| Cat | *Cryptosporidium* spp. | *Blastocystis* spp. |
| Dog Cat #1 | *Cryptosporidium* spp. | *Blastocystis* spp. |
| | *Cryptosporidium* spp. | |

Each row shows the parasite (s) found in the infected pet (s) and the respective owner.

Participants and pets showing the same parasite genus are indicated in shadow.

[a]Identified by Sanger sequencing.

Studies in vulnerable groups in Colombia have shown intestinal parasite prevalences up to 100%; for instance, 14.5% in adult population (19–48 years-old) [36], school children from urban areas 48%- 97% [37, 38] rural children 71%- 100% [38, 39], indigenous population 79–84% [40, 41] and pregnant women 41% [42]. Although this variability may be influenced by geographical area and the detection methods used, the university students seem to be also highly affected by parasites, moreover this population has the highest parasite prevalence when compared to similar studies worldwide [28–31, 34, 43]. Overall, ten different types of protozoa were found in the student´s fecal samples whereas geohelminths were not found.

Intestinal protozoans were also more prevalent in young adult populations from Ethiopia, Nigeria, Iran and Gaza [28, 31, 34, 43], these cases have been attributed to environmental

**Table 6. Prevalence of intestinal parasites in several published studies and the present work.**

| Study/ Location | Sample size | Age (mean) | Total prevalence | Parasite prevalence | Diagnostic method | Associations/Findings |
|---|---|---|---|---|---|---|
| Derso *et al.* 2021 [28] **Ethiopia** | 6244 | 18–35 y (21) | 45.6%(2850) | *Entamoeba* complex 20.3%<br>*G. intestinalis* 8.2%<br>*A. lumbricoides* 7.4%<br>Hookworm 5.2%,<br>*Taenia* sp. 1.4%, other 3.2% | Wet mount | Gender (males) |
| Ayele *et al.* 2019 [34] **Ethiopia** | 483 | 16–35 y (22) | 28.9% (140) | *Entamoeba* complex 19.7%<br>*G. intestinalis* 9.3% | Wet mount Formol ether | Marital status Rural residence University stay |
| Afolabi *et al.* **2016** [29] **Nigeria** | 300 | 10–50 y (25) | 13.3% (40) | *Entamoeba* complex 2.7%<br>*A. lumbricoides* 3.7%<br>Hookworm 2.3%<br>*E. vermicularis* 1.7%<br>*T. trichiura*, *G. intestinalis*,<br>*S. stercoralis*, 0.7% each<br>*S. mansoni* 1.0% | Flotation technique | Toilet type Feeding habits |
| **Khanum *et al.* 2013** [30] **Bangladesh** | 350 Staff, Student teacher | ND | 23.1% 20.4% 46/225 Students | *Entamoeba* complex 4.9%<br>*G.intestinalis* 3.7%<br>*A.lumbricoides* 11.1%<br>*T. trichiura* 3.4% | Wet mount Formol ether | Gender (female) |
| Fallahi *et al.* **2016** [31] **Iran** | 310 | 20–25 y (mainly) | 11.9% (37) | *Blastocystis* spp.4.5%<br>*G. intestinalis* 3.5%<br>*E. coli* 2.3%, *H. nana* 1.3%<br>*A. lumbricoides* 0.6%<br>*Entamoeba* complex 0.3%<br>Other protozoa 1.2% | Wet mount Formalin-ether Sheather Trichrome Modified ZN staining | Academic major |
| Al-Hindi, A *et al.* 2019 [43] **Gaza** | 305 Female | 18–22 y | 20.6% (63) | *Entamoeba* complex 7.5%<br>*G. intestinalis* 4.9%<br>*Blastocystis* 3.9% *E. coli* 2.6%<br>*D. fragilis* 1%<br>*A. lumbricoides* 0.3% | Formal-ether | NR |
| Ayala et al. 1972 [33] **Colombia** | 79 | ND | 81% (64) | *E. nana* 44%, *E. coli* 33%<br>*T. trichiura* 32%,<br>*A. lumbricoides* 18%<br>*G. intestinalis* 17%<br>*Entamoeba* complex 10%<br>Hookworm 9%, other 16% | Wet mount | NR |
| Ospina *et al.* 2006 [32] **Colombia** | 260 | 16–30 y | 83.8% (216) | *E. nana* 78.8%,<br>*Blastocystis* spp. 61.9%<br>*Entamoeba* complex 24.7%<br>*A. lumbricoides* 1.4%<br>*I. bütschlii* 2.8% | Wet mount | Street food consumption (fruit, juice) |

(*Continued*)

**Table 6.** (Continued)

| Study/ Location | Sample size | Age (mean) | Total prevalence | Parasite prevalence | Diagnostic method | Associations/Findings |
|---|---|---|---|---|---|---|
| Present study **Colombia** | 139 | 18–40 y (24) | 74.8% (104) | *Blastocystis* spp. 59.7% | Wet mount Flotation Trichrome Kinyoun stain PCR | Social assistance enrolment, *Blastocystis* and fruit consumption |
| | | | | *Cryptosporidium* 24.5% | | |
| | | | | *E. nana* 13.6%, *E. coli* 5% | | |
| | | | | *E. dispar/E. moshkovskii* 7.8% | | |
| | | | | Other protozoa 4.2% | | |

ND: no data, NR: not reported, y: years *E. vermicularis*: *Enterobius vermicularis*; *S. stercoralis*: *Strongyloides stercoralis*; *S. mansoni*: *Schistosoma mansoni*; *H. nana*: *Hymenolepis nana*.

contamination, for instance in Nigeria parasite infections were associated with feeding habits and toilet type [29]. This agrees with previous studies in Colombia in university students and other populations [32, 33, 42] where prevailing parasite infections may be linked to the ingestion of food and water contaminated with protozoa [44]. A recent review has reported a high rate of contamination in unwashed vegetables and fruits ranging from 3 to 49% [45]. Although less frequent, STH such as *A. lumbricoides*, *T. trichiura* and hookworm have been found in young adults (Table 6), however these infections are more commonly found in pre-school and school aged children from endemic areas. In this work, helminths were assessed by three microscopy methods, but they were not detected. Overall, it has been suggested that conventional methods might underestimate the true prevalence of STH infections.

In this study, both microscopy-based techniques and PCR for protozoa were used for parasite detection. Similar to previous studies, we found low level of concordance between both conventional microscopy and PCR, as judged by the kappa index [37]. Comparison of several microscopy techniques showed that trichrome stain was able to detect most parasites, whereas zinc sulphate flotation technique was slightly better than the wet mount, and it was not suitable for *Blastocystis* spp. as the chemical seems to induce damage of the parasite´s membranes [46, 47]. The combination of microscopy techniques used in this study presumptively detected 8% of pathogenic protozoa and no detection of *Cryptosporidium* spp. was reported using a modified acid-fast staining. However, PCR substantially improved parasite diagnosis by detecting *Cryptosporidium* spp. and *Blastocystis* spp. and allowing to differentiate *E. histolytica* from *E. dispar* and *E. moshkovskii* in the *Entamoeba* complex. Nevertheless, the only two cases of *G. intestinalis* reported by microscopy could not been confirmed by PCR, this may be due to several reasons such as sample inhibitors or incomplete cyst rupture. In this work, the most frequent pathogen was *Cryptosporidium* spp., and the most frequent commensal was *Blastocystis* spp. followed by *E. nana*. Comparison with other studies in young adults was limited by the detection methods (the others mainly based on microscopy) however, there was an agreement on the finding of *Blastocystis* spp., *E. nana* and *Entamoeba* complex [32] with exception of those in African countries where *Blastocystis* spp. was not reported [28, 29, 34]. For instance, in Ethiopia, the most frequent parasites reported in university students were *Entamoeba* complex (19.7–20.3%) and *G. intestinalis* (8.2–9.3%). Also, one study in Gaza mainly found *Entamoeba* complex (7.5%) and *G. intestinalis* (4.9%), in contrast with a work conducted in Bangladesh reporting *A. lumbricoides* (11.1%) followed by *Entamoeba* complex (4.9%) and *G intestinalis* (3.7%). In another study in Iran the prevailing parasites were *Blastocystis* spp. (4.5%) and *G. intestinalis* (3.5%). Previous studies in Colombia agreed with a high frequency of *E. nana* (44–78.8%) followed by *Blastocystis* spp. (61.9%) or *E. coli* (33%) [32, 33]. Only one

study from Iran included microscopy testing for *Cryptosporidium* sp. but cases were not found [31].

*Blastocystis* infections are highly prevalent globally, with frequencies up to 100% in some regions [48, 49]. In this survey, 59.7% of students were found infected with *Blastocystis* spp. which agrees with an earlier study in Colombia showing a colonization rate in students of 61.9% [32] and studies by Perez *et al.* and Guangorena *et al.* in Mexican students where *Blastocystis* spp. prevalence was 47% and 53% respectively, using microscopy and molecular approaches [15, 35]. Although *Blastocystis* spp. is considered a colonizer it may have a role in modification of gut microbiota and subtype specific effects on the individuals. It has been suggested that dysbiosis of gastrointestinal microbiota may also lead to chronic infections due to *Blastocystis* spp. but its impact on health is still matter of study [35]. *Blastocystis* spp. is also a marker of environmental contamination, and transmission has been associated with contaminated food or water and animal reservoirs, particularly the cattle [16]. Here, *Blastocystis* spp. infection was associated with fruit consumption, which is in agreement with findings by Ospina *et al.* where this parasite was associated with consumption of fruits, juices and salads [32]. Although most participants (92%) reported fruit and vegetable washing, judged by the results, this practice was not performed regularly, or contaminated water was used to wash the produce.

The second most prevalent parasite and the main pathogen found in this study was *Cryptosporidium* spp., a well-known pathogen found in surface water and fresh vegetables, is a main etiological agent of foodborne and waterborne outbreaks worldwide [44]. *Cryptosporidium* spp. is a highly infectious parasite with a low infection dose of 10 or even less than 10 oocysts and representing high risk for consumers [50, 51]. The main transmission route is drinking of untreated or contaminated water but various infection routes have been identified including livestock and person to person contact, foodborne transmission and contact with pets [52]. Outbreaks have been also reported in young veterinary students which are considered a high-risk population [53, 54]. In this work, most participants reported drinking regular tap water and 66.2% owned at least one pet. Interestingly, 24% of the participants were positive for *Cryptosporidium* spp. and all the cases were detected only by PCR. This is in contrast with studies performed in young adults in Iran [31] and vulnerable groups in Colombia, Brazil and Venezuela where *Cryptosporidium* spp. was not detected [42, 55, 56]. *Cryptosporidium* prevalence in this survey was substantially higher compared with the prevalence estimated for the country (7.8%) and studies using PCR in several bioregions in Colombia and Cuba which reported infection rates ranging from 0% to 10.5% [57, 58] but similar to one study in Colombia by Bryan *et al.* reporting 19.4% in an urban community [38]. *Cryptosporidium* spp. has been found in school children (2.4% to 9.8%) from Colombia with reports of *C. hominis* and *C. meleagridis* [37, 59, 60]. Globally, the estimated prevalence of *Cryptosporidium* sp. was 7.6% (95% CI 6.9–8.5), with the highest estimated prevalence of 69,6% in Mexico [11]. Nevertheless, these data can be still substantially underestimated as most studies in the developing world are only performed using microscopy examination. In this work, most participants with *Cryptosporidium* infection (21/34, 61.8%) were asymptomatic at the time of sampling. *Cryptosporidium* infections are mainly asymptomatic and self-limited but in young and immunosuppressed individuals it can cause acute watery diarrhea and it has been also associated with colon cancer [61, 62]. Moreover, colon cancer patients were found positive for *Cryptosporidium* sp. using microscopy (32.5%), ELISA (42.5%) and PCR (47.5%) showing significantly higher risk for infection when compared to the control group [62]. Long-term sequelae have been also described in immunocompetent individuals depending on the immune and nutritional status of hosts and parasite species, adaptation and virulence [63]. Several outbreaks have been documented in children [60] and immunocompromised patients,

moreover an outbreak affecting also immunocompetent adults was reported in French Guiana and linked to tap water consumption [64]. *C. hominis* is an etiological agent in outbreaks being clinically more severe and ease of transmission, particularly, higher virulence has been attributed to subtype IbA10G2 [63, 64].

The zoonotic pathogen, *G. intestinalis* was less frequent (1.4%), this in agreement with studies in university students from Colombia, Asian and African countries with rates ranging from 0% to 9.3% [28–30, 32]. By contrast, other population groups in Colombia have shown higher *G. intestinalis* rates such as 19.4% in adults, 39%-45% in children living in rural areas, 10.6–48% indigenous people and 28% in pregnant women [36, 40–42, 65].

The potential pathogen *D. fragilis* was only found in one case using trichrome stain which is not a microscopy routine test. *D. fragilis* detection by several stain methods is challenging and little is known about its impact in low- and middle-income countries. Studies in Latin America have shown prevalences up to 40% depending on the region and population group [66]. For instance, *D. fragilis* has been found in 15% of asymptomatic individuals [55] and 10.3% in children from Brazil [66] and 40.4% in a rural community in Venezuela [56].

Regarding the second most common commensal, *E. nana*, our results agree with the overall in healthy population (13.9%) [67] although is lower than the reported by Ospina *et al*. in a young population 78.8% [32]. *E. nana* is considered an indicator of fecal contamination in food and water, and it has been found in banknotes. Although no pathogenic associations are known its role as modulator of the immune response has not been rule out [67].

None of the cysts presumptively identified as *E. histolytica/E.dispar/E. moshkovskii* by microscopy were *E. histolytica*, this is similar to other studies in Colombia and other countries using PCR to detect *E. histolytica* and reporting very little or no detection of the pathogen (0%- 0.39%) [37, 57, 58, 68]. Worldwide trends reveal a decline in *Entamoeba* infections although a high burden is still seen in some age groups and low-income regions [69]. Prevalence of *E. histolytica* may vary in the most vulnerable populations however, studies only based on microscopy examination often result in overestimation of this protozoan. A main advantage of molecular detection is to distinguish *E. histolytica* from commensal species in the *Entamoeba* complex allowing a better understanding of its epidemiology. The commensal amoeba *E. dispar* seems to be more prevalent than *E. histolytica* with 12% prevalence worldwide [69], likewise in this study, we found *E. dispar* rather than *E. histolytica*. Although *E. dispar* has been considered a noninvasive amoeba, studies by Vilela *et al*. have shown that virulence factors may be selectively expressed in South American strains leading to clinical disease [70].

Polyparasitism has been associated with higher exposure to contaminated environment, high level of environmental contamination and multiple routes of transmission. Here, one third of the parasitized participants had more than one parasite, this is slightly lower than the previously reported in a similar population in Colombia (41.7%) but with similar rate of more than two infections (14% vs 17.4%) [32]. Multiple parasite infections may indicate alterations on the immune response or nutritional status and increased susceptibility to re-infection, severe disease and other host infections. Overall, student young population showed lower level of polyparasitism (37.5%) compared to school children in Colombia (83%) [65] and other vulnerable groups (52–61%) [36, 40]. We found that pet owners were protected from polyparasitism, which has been attributed to activation of the immune response.

Pathogens of public health concern such as *Cryptosporidium* spp., *Giardia* spp., hookworm and *T. gondii* were found in pets, moreover *Cryptosporidium* spp. and *Giardia* spp. were found in both humans and pets. Furthermore, the estimated prevalence of *Cryptosporidium* spp. in people with animal contact was 18%, similar to that of those living in non-urban areas [11]. Remarkably, in this survey *Cryptosporidium* spp. was also highly prevalent in companion animals, this agrees with estimates suggesting that 8% of *Cryptosporidium* infections are

transmitted by pet contact [52]. Studies in companion animals have shown that *C. canis* and *C. felis* are frequently found in dogs and cats, whereas *C. parvum*, the most common species in humans, has been found in a wide range of hosts [50]. *C hominis* and *C. parvum* are the most predominant in humans followed by *C. canis* and *C. felis* which are host specific species [52, 60]. We identified *C. felis* in one participant who owned three pets, all of them positive for *Cryptosporidium* spp. This finding of a host specific *Cryptosporidium* spp. may support transmission between owners and their companion animals. One participant with *C. parvum* owned two pets but fecal samples from those pets were not submitted for testing.

The prevalence of *Cryptosporidium* spp. in animal surveys is also highly variable and depends on the geographical area, climate, and environmental contamination. Animals acquire parasites from exposure to untreated water, raw meat feeding or environmental contamination with oocysts [71–73]. We reported *Cryptosporidium* spp. in half of the dogs and cats. Other studies in Latin America have shown the same trend, however the infection rate in our study was substantially higher compared to studies using different detection methods and ranging from 4% to 24.5% [71, 74, 75]. A recent review in cats reported a worldwide prevalence of *Cryptosporidium* sp. of 6% [76] whereas in Brazil and Colombia the prevalence in cats was 11.1% and 13% respectively [74, 77].

Interestingly, we found *T. gondii* oocysts in one domestic cat (6.6%). *T. gondii* is a zoonotic protozoa and its prevalence in humans has been correlated to oocyst contamination of environments and infected stray cats. Several studies in Colombia have shown prevalence of oocyst shedding in cats ranging from 0 to 66% [78–80] with fluctuations depending on the region, cat population and low or sporadic oocyst shedding. A recent study using DNA detection by PCR found 17.8% *T. gondii* positive fecal samples from cats [79]. Oocyst shedding is considered of epidemiological significance for the dynamics and toxoplasmosis transmission in animals and humans, this highlights the need to further investigate *T. gondii* infections in domestic cats from our region to control the spread.

The proximity among animals and humans and particularly pets in children and young adults represents a potential risk for infection. We observed that several animals in a household are highly likely to be infected with the same parasites suggesting either high transmission and/or environmental contamination. Several factors related to the human and animal habits, the parasite and the environmental conditions may contribute to the high rate of *Cryptosporidium* spp. infections seen in this survey. Studies in domestic animals have reported a prevalence ranging from 6.7% to 41.6% [76, 81], a recent study has reported a pool prevalence of 18% in Latin America [82], 20.3% in pets and 19.9% in livestock, the latter being considered so far as the main parasite reservoir [82]. In Colombia, the *Cryptosporidium* prevalence in livestock has been reported to be from 13 to 26.6% [83, 84].

Overall, the higher susceptibility of young adults to protozoan parasites and particularly *Cryptosporidium* spp. may be associated with several routes of transmission however, this study points out a relevant level of environmental contamination and the potential role of pets as source of infection. For a One Health approach, it is paramount to monitor the potential sources of contamination and their contribution to human infection; this includes assessment of drinking water and animal health. Previous studies in Latin America have found *Cryptosporidium* spp. in raw and drinking water [85–88] as well as outbreaks affecting immunocompetent and immunocompromised individuals [64]. Interaction of humans and animals has been growing and becoming closer in recent years increasing the potential for zoonotic transmission, and spread of pathogens such as *Cryptosporidium* spp. and *G. intestinalis*. As suggested by others, surveillance requires molecular detection to understand the real epidemiology as microscopy has poor sensitivity and underestimates the real prevalence of infection [89]. Further characterization of circulating genotypes, virulence and adaptation mechanisms to

environmental conditions is needed to understand transmission routes and to focus interventions on human, animal, and environmental health. One limiting factor of this study was the small animal sample, further pet surveys should also assess pet ownership practices, pet habits and behavioural factors.

## Conclusions

Prevalence of protozoa in university students was remarkably high suggesting that this group is a relevant susceptible population for surveillance as short- and long-term effects remain to be determined. More attention should be given to young adults as many of them are in school to work transition and underlying morbidity or coinfections may be worsening the outcome of parasitic infections. Particular habits and conditions in this population may increase the exposure to some routes of transmission leading to the selective spread of some parasite types. Molecular subtyping would be useful to determine the main transmission route and the virulence potential or effects of *Cryptosporidium* spp. and the *Blastocystis* host interactions.

Control strategies to improve student´s health and to prevent co-infections should be focused on education programs for better identification of contamination sources for both humans and pets and to reinforce food safety to reduce exposure. Boiling of drinking water, well-cooked meals and proper washing of fruit/vegetables should be a basic measurement to control *Cryptosporidium* spp. as the oocysts are resistant to most disinfection treatments. Our findings also encourage veterinary care and control of parasitic infections in domiciled animals, in addition to implementation of better diagnostics and prophylactic programs in our region. This study has also shown that molecular detection for *Cryptosporidium* spp. is a need for diagnosis and surveillance.

## Supporting information

**S1 Appendix. Gel images.**
(PDF)

**S2 Appendix. GenBank accession numbers.**
(PDF)

**S1 Raw images. Raw gel images.**
(PDF)

## Acknowledgments

We are sincerely grateful to David Summerhold, Guillermo Nevado and Anthony Bolaños for their help with the recruitment of participants and general technical procedures. We also thank Meleny Ramirez and Claudia Auseche (Department of Microbiology, Universidad del Valle) for technical assistance with the microscopy work.

## Author Contributions

**Conceptualization:** Maria del Pilar Crespo-Ortiz.

**Data curation:** Caterine Potes-Morales.

**Formal analysis:** Caterine Potes-Morales, Maria del Pilar Crespo-Ortiz.

**Funding acquisition:** Maria del Pilar Crespo-Ortiz.

**Investigation:** Caterine Potes-Morales.

**Methodology:** Caterine Potes-Morales, Maria del Pilar Crespo-Ortiz.

**Project administration:** Maria del Pilar Crespo-Ortiz.

**Writing – original draft:** Maria del Pilar Crespo-Ortiz.

**Writing – review & editing:** Caterine Potes-Morales.

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
