## [Decision Letter · Decision Letter 0]

6 Sep 2022

PONE-D-22-21617Molecular diagnosis of intestinal protozoa in young adults and their pets in Colombia, South AmericaPLOS ONE

Dear Dr. Maria Crespo-Ortiz, Thank you for submitting your manuscript to PLOS ONE. After careful consideration, we feel that it has merit but does not fully meet PLOS ONE’s publication criteria as it currently stands. Therefore, we invite you to submit a revised version of the manuscript that addresses the points raised during the review process. Please submit your revised manuscript by October 21, 2022. If you will need more time than this to complete your revisions, please reply to this message or contact the journal office at plosone@plos.org. Please include the following items when submitting your revised manuscript:A rebuttal letter that responds to each point raised by the academic editor and reviewer(s). You should upload this letter as a separate file labeled 'Response to Reviewers'.A marked-up copy of your manuscript that highlights changes made to the original version. You should upload this as a separate file labeled 'Revised Manuscript with Track Changes'.An unmarked version of your revised paper without tracked changes. You should upload this as a separate file labeled 'Manuscript'.

We look forward to receiving your revised manuscript.

Kind regards,

Saeed El-Ashram

Academic Editor

PLOS ONE

Journal Requirements:

"We also acknowledge Meleny Ramirez and Claudia Auseche (Department of Microbiology, University del Valle) for assistance with the microscopy work."

"This research was financially supported by the Internal Grant Scheme 2020 (CI1920) awarded to CPM and MPC by the Universidad del Valle, Cali, Colombia. The funder had no role in study design, data collection and analysis, decision to publish, or preparation of the manuscript."

3. We note that Figure 1 in your submission contain map images which may be copyrighted. All PLOS content is published under the Creative Commons Attribution License (CC BY 4.0), which means that the manuscript, images, and Supporting Information files will be freely available online, and any third party is permitted to access, download, copy, distribute, and use these materials in any way, even commercially, with proper attribution. For these reasons, we cannot publish previously copyrighted maps or satellite images created using proprietary data, such as Google software (Google Maps, Street View, and Earth). For more information, see our copyright guidelines: http://journals.plos.org/plosone/s/licenses-and-copyright.

Reviewers' comments:

Reviewer's Responses to Questions

**Comments to the Author**

1. Is the manuscript technically sound, and do the data support the conclusions?

Reviewer #1: Partly

Reviewer #2: Yes

Reviewer #3: Yes

2. Has the statistical analysis been performed appropriately and rigorously? 

Reviewer #1: Yes

Reviewer #2: Yes

Reviewer #3: Yes

3. Have the authors made all data underlying the findings in their manuscript fully available?

Reviewer #1: Yes

Reviewer #2: Yes

Reviewer #3: Yes

4. Is the manuscript presented in an intelligible fashion and written in standard English?

Reviewer #1: Yes

Reviewer #2: Yes

Reviewer #3: Yes

5. Review Comments to the Author

Reviewer #1: The Graph (s) presentation is missing, some selective addition of tables. There may be inclusion of broad area study including various municipalities and districts and more study focus on old age subjects or host owners and immuno-compromised persons.

Reviewer #2: Abstract

Line 46: Suggest removing ‘highly contaminated environment’ and replacing with a phrase that indicates interaction and exposure to an infected animal and/or infected animal’s environment.

Please include genus and species if possible.

Introduction

Line 55-58: References needed

Line 58: Migration of wild animals? Please specify.

Line 64: Please describe what Disability Adjusted Life Years is and how it is relevant.

Line 74: Describe why the range is so large for Dientamoeba infections

Line 79: High variability due to what factors?

Materials and Methods:

Line 111: Selection criteria for students? Describe clinical variables. Was any data excluded for a particular student? If so, please justify.

Consider including references for microscopy methods that have been described as appropriate.

Table 1. Should be thermocycling, not termocycling.

Statistical Analysis: Did you evaluate any interactions between variables?

Results

Line 87: Include n number consistently

Table 2. Include total n number for positive and negative. Describe social assistance program since this is the only variable where significance was observed. Remove the footnote about having more than one pet or clarifying for the other similar variables as well. Consistency is key.

Again, please be consistent with n numbers, etc. throughout the entire manuscript.

Table 3. Why does the n number differ across the floats and staining technique? Please describe how you differentiated between the protozoa based on morphology or include reference. Please include zeros or NA in the table if you did not detect or evaluate.

Line 250: How did you determine overall prevalence? Clarify please. Include n number for reference. Did you have more false positives with microscopy or PCR? Do you think the DNA extraction method used reduced detection of Endolimax nana by PCR?

Line 263-264: Split into two sentences.

Table 5. Provide more information to help reader interpret this data. It is not straightforward.

Supplementary Gel Image 2. Is there another band ~105-110bp?

Reviewer #3: It was an extensive search. It is remarkable that parasites, which constitute an important public health problem, have been identified. I also consider the coexistence of several parasites to be a valuable finding. It is important for me to answer the parts I have explained below. I also think that some corrections will make the article more fluent and understandable.

The first thing that caught my attention is why T. gondii infections, one of the serious zoonotic infections, are ignored. Should have been included in this study in some way. However, Echinococcus gronulosus could also be investigated. should not be restricted to protozoal infections only.

Its importance in humans and animals should be considered in more detail. As a result, a unilateral infection does not occur.

The charts are nice, but a separate colored chart with co-infections would be more effective. It will be more understandable if the work in a more graphic style is strengthened.

6. PLOS authors have the option to publish the peer review history of their article (what does this mean?). If published, this will include your full peer review and any attached files.

Reviewer #1: No

Reviewer #2: No

Reviewer #3: No

---

## [Author Response · Author response to Decision Letter 0]

18 Dec 2022

We have attached a file with the response to each point raised by the reviewers.

---

## [Decision Letter · Decision Letter 1]

5 Feb 2023

PONE-D-22-21617R1Molecular diagnosis of intestinal protozoa in young adults and their pets in Colombia, South AmericaPLOS ONE

Dear Dr. Maria Crespo-Ortiz,

Thank you for submitting your manuscript to PLOS ONE. After careful consideration, we feel that it has merit but does not fully meet PLOS ONE’s publication criteria as it currently stands. Therefore, we invite you to submit a revised version of the manuscript that addresses the points raised during the review process.

We look forward to receiving your revised manuscript.

Kind regards,

Saeed El-Ashram

Academic Editor

PLOS ONE

Additional Editor Comments:

Please respond to reviewer number one.

Reviewers' comments:

Reviewer's Responses to Questions

**Comments to the Author**

1. If the authors have adequately addressed your comments raised in a previous round of review and you feel that this manuscript is now acceptable for publication, you may indicate that here to bypass the “Comments to the Author” section, enter your conflict of interest statement in the “Confidential to Editor” section, and submit your "Accept" recommendation.

Reviewer #1: All comments have been addressed

Reviewer #3: All comments have been addressed

Reviewer #4: All comments have been addressed

2. Is the manuscript technically sound, and do the data support the conclusions?

Reviewer #1: No

Reviewer #3: Yes

Reviewer #4: Yes

3. Has the statistical analysis been performed appropriately and rigorously? 

Reviewer #1: No

Reviewer #3: Yes

Reviewer #4: Yes

4. Have the authors made all data underlying the findings in their manuscript fully available?

Reviewer #1: No

Reviewer #3: Yes

Reviewer #4: Yes

5. Is the manuscript presented in an intelligible fashion and written in standard English?

Reviewer #1: Yes

Reviewer #3: Yes

Reviewer #4: Yes

6. Review Comments to the Author

Reviewer #1: A thorough study to be conducted. Missing graph and proper discussion in the manuscript with meager latest references

Reviewer #3: It was good field work. It is remarkable that the interactions between humans and animals in the Columbia region are evaluated in terms of disease. I see you have made the corrections.

Reviewer #4: After the revision of this MS. IT noticed that authors presented a well organized data and addressed all the required points. This MS is now in acceptable form

7. PLOS authors have the option to publish the peer review history of their article (what does this mean?). If published, this will include your full peer review and any attached files.

Reviewer #1: No

Reviewer #3: No

Reviewer #4: No

<quillbot-extension-portal></quillbot-extension-portal>

---

## [Author Response · Author response to Decision Letter 1]

22 Feb 2023

Dear Editor,

Thank you for your helpful comments regarding our manuscript: “Molecular diagnosis of intestinal protozoa in young adults and their pets in Colombia, South America”. 

Please find below a point-by-point response to editors and reviewer. 

To Editor comments:     

Please respond to reviewer number one

Authors: We have carefully read the reviewer # 1´s comments. The observations are too general, but we have added some information expecting to fulfill all the requirements and suggestions. We have observed that after the first round of evaluation all the reviewers agreed with questions #1 to #5, however, in the last assessment the reviewer #1 changed the answers in questions #2, #3 and #4.

Regarding this we have shown in detail all methods and the supporting information, including the raw gel images. The data analysis was conducted using SPSS v 27 and results were also verified in Epi info v 7.2.5. In the manuscript (Statistical analysis section) we have also included the multiple logistic regression analysis which further support our results. 

To Reviewer comments:  

Reviewer #1: A thorough study to be conducted. Missing graph and proper discussion with meager latest references.

Authors: We have added three figures highlighting our results (Fig 1) and showing in more detail our data and the analysis performed (Fig 2 and 3). To improve the discussion section, we have included Table 6 comparing our data with published studies in similar young adult populations worldwide. 

The references were revised and updated as far as not many published studies have been conducted in student young adults. 

As we stated before we agree that studies in other population groups should be included for surveillance particularly for Cryptosporidium, those studies may correspond to future research. We have provided the starting point for this research and reinforce the need for molecular testing. The presented work has focused on young adult population because the data on intestinal parasites is scarce, and the effects of parasite infection are unknown. In our latest literature review we could not find a study with a similar approach in the young adult population and exploring the involvement of animal exposure.

---

## [Editor Report · Decision Letter 2]

28 Feb 2023

PONE-D-22-21617R2Molecular diagnosis of intestinal protozoa in young adults and their pets in Colombia, South AmericaPLOS ONE

Dear Dr. Maria Crespo-Ortiz,

Thank you for submitting your manuscript to PLOS ONE. After careful consideration, we feel that it has merit but does not fully meet PLOS ONE’s publication criteria as it currently stands. Therefore, we invite you to submit a revised version of the manuscript that addresses the points raised during the review process.

ACADEMIC EDITOR: Please write a proper discussion and include more recent references. Delete this map and replace it with a more accurate one with a scale. Avoid using the Google map.

We look forward to receiving your revised manuscript.

Kind regards,

Saeed El-Ashram

Academic Editor

PLOS ONE

Journal Requirements:

<quillbot-extension-portal></quillbot-extension-portal>

---

## [Author Response · Author response to Decision Letter 2]

9 Mar 2023

Dear Academic Editor, as per indicated in the cover letter and response to editor, we have followed all the suggestions needed to have our paper published. Thanks very much.

---

## [Editor Report · Decision Letter 3]

20 Mar 2023

Molecular diagnosis of intestinal protozoa in young adults and their pets in Colombia, South America

PONE-D-22-21617R3

Dear Dr. Maria,

We’re pleased to inform you that your manuscript has been judged scientifically suitable for publication and will be formally accepted for publication once it meets all outstanding technical requirements.

Kind regards,

Saeed El-Ashram

Academic Editor

PLOS ONE

Additional Editor Comments (optional):

Reviewers' comments:

<quillbot-extension-portal></quillbot-extension-portal>

---

## [Editor Report · Acceptance letter]

11 May 2023

PONE-D-22-21617R3 

Molecular diagnosis of intestinal protozoa in young adults and their pets in Colombia, South America 

Dear Dr. Crespo-Ortiz:

I'm pleased to inform you that your manuscript has been deemed suitable for publication in PLOS ONE. Congratulations! Your manuscript is now with our production department. 

Kind regards, 

on behalf of

Professor Saeed El-Ashram 

Academic Editor

PLOS ONE